# Effect of Post-Hatch Heat-Treatment in Heat-Stressed Transylvanian Naked Neck Chicken

**DOI:** 10.3390/ani11061575

**Published:** 2021-05-27

**Authors:** Roland Tóth, Nikolett Tokodyné Szabadi, Bence Lázár, Kitti Buda, Barbara Végi, Judit Barna, Eszter Patakiné Várkonyi, Krisztina Liptói, Bertrand Pain, Elen Gócza

**Affiliations:** 1Animal Biotechnology Department (AB), Institute of Genetics and Biotechnology (GBI), Hungarian University of Agriculture and Life Sciences, 2100 Gödöllő, Hungary; toth.roland.imre@uni-mate.hu (R.T.); tokodyne.szabadi.nikolett@uni-mate.hu (N.T.S.); lazar.bence@uni-mate.hu (B.L.); 2Institute for Farm Animal Gene Conservation, National Centre for Biodiversity and Gene Conservation, 2100 Gödöllő, Hungary; buda.kitti@nbgk.hu (K.B.); vegi.barbara@nbgk.hu (B.V.); barna.judit@nbgk.hu (J.B.); varkonyi.eszter@nbgk.hu (E.P.V.); liptoi.krisztina@nbgk.hu (K.L.); 3Stem-Cell and Brain Research Institute, U1208 INSERM, USC1361 INRA, 69675 Bron, France; bertrand.pain@inserm.fr

**Keywords:** poultry, heat-treatment, heat-shock proteins, heat-shock factors, climate change

## Abstract

**Simple Summary:**

Heat stress due to high environmental temperature negatively influences animal productivity. Extensive studies have been carried out to evaluate the mechanisms of heat stress in chickens. It was shown that the expression level of heat-shock factors (HSFs) and heat-shock proteins (HSPs) were affected. Tissue-specific responses to the thermal challenge were also found in the heart, liver and muscle. Our study examined the changes in primary production parameters and four heat-shock factor and two heat-shock protein expression profiles in chicken gonads. In the first experiment, 24 h after hatching, 80 Transylvanian Naked Neck chickens were heat-treated at 38.5 °C ambient temperature with 60% humidity for 12 h. In this experiment, we studied the primary productivity parameters of matured chickens after the performed heat stress. In the second experiment, the heat treatment was the same, and we examined the expression pattern of heat-shock factors and heat-shock proteins in the control and treated gonads. We collected the samples immediately after the heat-treatment in case of half of the treated and control group. We found a significant difference in egg production, and increased expression level of *HSP90* and *HSF4* in heat-treated female gonads.

**Abstract:**

Although numerous studies reported the effects of heat stress in chickens, it was not investigated in the Transylvanian Naked Neck breed. In our research, Transylvanian Naked Neck chickens, 24 h after hatching, were heat-treated at 38.5 °C for 12 h. We compared the control and heat-treated adult chickens’ productivity parameters following 12 weeks of heat-stress at 30 °C. We found that the heat-treated layers had significantly higher egg production in heat stress, but in cockerels, the sperm quality did not differ significantly between the two groups. To detect the effect of heat-treatment on a molecular level, the expression of two heat-shock proteins and four heat-shock factors were analysed in the gonads of control and heat-treated chickens. We found that the expression level of *HSP90* and *HSF4* increased significantly in heat-treated female chicken gonads. Still, in adult females, the expression of *HSF2* and *HSF3* were substantially lower compared to the control. In adult heat-treated males, the *HSP70*, *HSF1* and *HSF3* expression levels showed a significant increase in both gonads compared to the control. We think that the presented significant differences in egg production might be related to the increased expression level of *HSP90* and *HSF4* in heat-treated female gonads.

## 1. Introduction

One of the most important environmental factors is the ambient temperature and due to the recent changes in global climate the average temperature is rising. Agriculture is very sensitive to the climate variability and extreme weather [1]. An increasing number of articles were published in previous years about the effects of climate change in agriculture [2,3] and in animal husbandry [1,4,5,6]. Animal exposure to hot environments deleteriously affects their reproductive functions [7]. Chickens are homeothermic animals and their body temperature is maintained in the range of 41 to 42 °C [8]. Higher environmental temperature negatively impacts the feed intake, the reproductive function, the hatchability and the meat and egg production [9,10,11,12,13,14]. In chickens, the high ambient temperature affects their endocrine system, reproductive and egg-laying performance, too [15].

Many researchers examined the effect of thermal manipulation in chicken. Two types of thermal manipulation are known: during the incubation period or after hatching. According to Al-Rubika et al. [16], thermal manipulation during the late embryogenesis has not affected the hatchability, but the body weight of thermally manipulated embryos was higher than the control group. Walstra et al. [17] investigated the effect of temperature manipulation on the behaviour of layer chickens. The thermally manipulated embryos were incubated at 37.8 °C, but between the 14th and 18th embryonic day the embryos were exposed to 40 °C for 4 h. The manipulated chicks preferred the lower ambient temperature, but no effect of thermal manipulation on behaviour and performance were observed [17]. Rozenboim et al. (2007) [15] examined the heat stress at the 31st life week for 12 h per day in White Leghorn Laying hens. They checked the body temperature, egg production, egg weight, feed intake and plasma hormone levels. They found the egg production was significantly lower in the heat stressed group. The effect of thermal manipulation was not detected in case of body weight in broiler, but the manipulation induced the up-regulation of muscle grow factors and muscle marker genes [18]. Vinoth et al. [19] investigated the effect of thermal manipulation and thermal stress on *HSP* gene expression, DNA methylation in brain tissue of Naked Neck and Punjab Broiler-2 chicken breeds. In hot regions the Naked Neck gene is very common [20]. They found that the DNA methylation level was lower, and the gene expression was higher in case of heat stressed but not heat-treated chickens, compared with other experimental groups (non-heat-treated but heat stressed; heat-treated and not heat stressed; and heat-treated and stressed). Rajkumar and his colleagues [21] demonstrated, that the Naked Neck breeds have better growth performance in higher temperature than the normal siblings, and the Heterophil/Lymphocyte ratio was significantly lower, what is indicating that Naked Neck chickens were less stressed in higher temperature.

In case of heat stress, a lot of heat-shock proteins (HSPs) and heat-shock factors (HSFs) start to be expressed to protect the cells from the effect of heat stress [22,23]. According to their molecular weights six HSP families are known (small HSPs, HSP40, HSP60, HSP70, HSP90 and HSP100) [24]. HSP70 and HSP90 are highly conserved ATP-dependent molecular chaperons that are essential for the eucaryote systems in unstressed conditions [25]. Of the many HSPs, the HSP70 correlates the best with thermotolerance [26]. The HSP90 chaperone is present in bacteria and all of eucaryotes. HSP90 participates in collaboration with the HSP70 chaperon system in protein folding and activation [27,28,29]. Increased HSP70 expression was detected in Japanese quail’s myocardial tissue in case of isolation in darkness, loud noise and cold temperature [30]. These results denoted that the heat-shock proteins are expressing in different types of stress. Under stress condition in poultry most heat-shock factors are activated. In animals, four HSFs were known until 2018, when Saju and his colleagues found [31] the fifth HSF: HSF5 in *Danio rerio*. The HSF5 is essential for the spermatogenesis in zebrafish. Two isoforms of HSF1 was discovered in gonads and liver of Danio [32]. In chicken, the *HSF1*, *HSF2* and *HSF3* genes were isolated by cross-hybridization with mouse hsf1 cDNA probe [33]. *HSF1* was mapped to the 2nd chromosome, the *HSF2* to the 3rd, the *HSF3* to the 4th and the *HSF4* gene to the 11th chromosome in chicken [34]. Zhang and his colleagues [10] examined the effects of acute heat stress in two different Chinese chicken breeds. They found that with increasing heat treatment time, both HSF3 and HSP70 expression first decreased then showed a significant increase in both breeds. They found that the expression of HSF3 and HSP70 is species-specific and tissue-specific during heat treatment.

During heat stress different poultry breeds react differently. The broiler breeds are more sensitive than the local breeds. The results of heat tolerance indexes suggested that with age the local breeds easily overcome heat stress, while the other chickens become increasingly vulnerable. These results have been confirmed by the high mortality rate observed in the commercial stock under heat stress, while there was no mortality among the local chickens [35].

In our study we investigated the effect of heat treatment and heat stress on the egg production, sperm quality, heat-stress protein and heat-shock factor expression profile in the Transylvanian Naked Neck Hungarian chicken breed.

## 2. Materials and Methods

### 2.1. Animals

Eggs of Speckled Transylvanian Naked Neck (STN) chickens were from the National Centre for Biodiversity and Gene Conservation—Institute for Farm Animal Gene Conservation (NBGK-HGI), Gödöllő, Hungary. Hatching and heat treatment took place in the experimental hatchery, the comparative study under heat stress was done in the animal house of this institution.

### 2.2. Heat Treatment

The eggs were incubated in a MIDI F500S hatchery machine (Tárnok, Hungary) in the regular way. For the first 24 h after hatching, all 160 chicks were placed under an infrared lamp at 32 °C on absorbent paper litter with ad libitum starter feed and water. Then, 80 chicks were placed back to the hatcher for heat treatment. The temperature was set to 38.5 °C and the humidity was 60% for 12 h. Drinking water and feed were ad libitum inside. The control 80 chicks were kept at 32 °C. Then, in Experiment I the animals were kept and raised together with the control group, in Experiment II 15 treated and 15 control chicks were immediately sacrificed for RNA analysis.

### 2.3. Heat Stress

In Experiment I, 80–80 animals were in the treated and control groups. We randomly selected for the further reproductive biology examinations under heat stress 31–31 layers with 3–3 males in the 24th life week. Both of them were kept on a wood chips and zeolite mixture litter with ad libitum layer feed and water under 16-h lighting, with egg nests (10) and perches. Additionally, 10–10 roosters were placed in the same air space, in individual cages, for semen examinations, with ad libitum rooster feed and water. Ventilation was provided 10 min per hour. The average temperature in the pen at the height of the birds’ habitat was constantly 30 °C from 24th life week during 12 weeks for all treated and control animals.

### 2.4. Examination of Embryonic Abnormalities

In Experiment I, eggs were collected daily from the 24th life week and marked with group number and date and placed to the incubator once a week. Every week 20–20 eggs were applied for molecular biological examinations. The incubation was made the regular way (37.5 °C and 60% humidity). Eggs were candled on the 7th day of incubation. The ratio of fertile and infertile eggs was determined.

### 2.5. Semen Collection and Classification

Sperm-donor animals were selected on the basis of responsiveness to semen collection, followed by individual semen evaluation data. Semen collection was performed using Burrows and Quinn’s [36] dorso-abdominal massage technique twice a week for 3 months—from 23 weeks of age to 34 weeks of age—following a two-week training period. Semen was classified weekly for the following spermatological parameters:Volume (mL): determined with a pipette;Motility: determined by subjective estimation using a light microscope (Leica) on a scoring scale from 0 to 5 at 40× magnification. The test was always performed by the same experienced person;Concentration: determined with a spectrophotometer (Accucell IMV, France). At the beginning of the experiment, the instrument was calibrated. A concentration curve was established by comparing the spectrophotometer data of the samples in a dilution series with the concentrations determined using the Makler chamber;Type of morphological abnormalities and live/dead cell ratio: the study was performed using eosin–aniline blue vital staining [37].

### 2.6. Collection of Gonadal Tissues

In Experiment II, the chickens were euthanized by cervical dislocation after the treatment. The tissue samples were collected in sterile plastic dishes. Small pieces from each gonad were placed into RNAlater^TM^ Solution (Thermo Fisher Scientific, 145 Waltham, MA, USA). We collected thigh muscle samples from each chicken for DNA analysis. The samples were transferred into TRizol^TM^ reagent after two days. The samples were incubated in TRizol^TM^ (Thermo Fisher Scientific, 145 Waltham, MA, USA) for 10 min at room temperature than stored at −80 °C.

### 2.7. DNA Isolation and Sex Determination

The thigh muscle samples were digested using 0.1 % Proteinase-K Lysis Buffer solution and incubated at 55 °C for 3 hours. After the incubation, the Proteinase-K was inactivated at 99 °C for 10 min. Total DNA was extracted using the Phenol-chloroform DNA isolation protocol [19]. The isolated DNA was quantified by measuring the absorbance at 260 nm using a NanoDrop One Spectrophotometer (Thermo Fisher Scientific, 145 Waltham, MA, USA). The purity was assessed by determining the ratio of the absorbance at 260 and 280 nm.

The sex of treated and control chickens were determined using CHD1 (Chromosome Helicase DNA binding protein 1) primer set (Appendix A). The isolated DNA was diluted to 25 ng/µL for the PCR and gel electrophoresis. PCR was performed using MyTaq Red Mix (Thermo Fisher Scientific, 145 Waltham, MA, USA). A total of 13 µL of the reaction solution was used: 6.75 µL MyTaq Mix; 0.5 µL reverse CHD1 primer (10 µmol); 0.5 µL forward CHD1 primer (10 µmol); 4.25 µL sterile water; and finally, 1 µL DNA sample. The cycling parameters were 95 °C for 1 min. 28 cycles of 95 °C for 15 s followed 30 s at 48 °C and 72 °C for 10 sec. It was finally melting at 72 °C for 5 min. The PCR products were separated by electrophoresis, using 1.5 % agarose gel stained with ethidium bromide, at 100 V for 30 min. The DNA bands were visualized under UV illumination and photographed.

### 2.8. RNA Isolation, cDNA Writing, and Real-time qPCR

Total RNA extraction and purification from cells collected in the TRIzol Reagent was following the manufacturers’ protocol. The RNA was quantified by measuring the absorbance at 260 nm using a NanoDrop One Spectrophotometer (Thermo Fisher Scientific, 145 Waltham, MA, USA), and the purity was assessed by determining the ratio of the absorbance at 260 and 280 nm. Total RNA (15 μL) was reverse transcribed using a cDNA synthesis kit (High-Capacity cDNA Reverse Transcription Kit, Thermo Fisher Scientific, 145 Waltham, MA, USA). SYBR Green PCR master mix was applied for the qPCR as a double-stranded fluorescent DNA-specific dye according to the manufacturer’s instructions (Thermo Fisher Scientific, 145 Waltham, MA, USA). The primers used for real-time PCR are displayed in Appendix A. Amplification was carried out in a total volume of 15 μL containing Power SYBR Green PCR Master Mix (Thermo Fisher Scientific, 145 Waltham, MA, USA), forward and reverse primers (0.1 μg/μL), sterile water (Thermo Fisher Scientific, 145 Waltham, MA, USA) and 0.75 μL of cDNA. After an initial 10 min denaturation step at 95 °C, the reactions were cycled 40 times under the following parameters: 95 °C for 15 s, 60 °C for 40 s and 68°C for 20 s. Optical detection was carried out at 68 °C.

We tested the expression of two housekeeping genes (*GAPDH* and *ß-Actin*) [11]. We compared the average Ct values of *GAPDH* and *ß-Actin* in control and heat-treated left gonads. We decided to use *GAPDH* as we found lower Ct values in the case of *GAPDH*, and the standard deviation were higher using *ß-Actin* (Appendix A). There was no significant difference between the control and heat-treated samples (*p* = 0.523) comparing the average *GAPDH* Ct values. We used chicken embryonic fibroblast as a reference sample [11,38,39]. All reactions were performed in triplicate. Those qPCR measurements were used at the analysis where the reference sample (CEF) CT value on the plate has not significantly different from the average CEF CT value. From the collected gonads we performed qPCR to analyse the expression profile of two heat-shock protein (*HSP70*, *HSP90*) genes and four heat-stress factor (*HSF1*, *HSF2*, *HSF3*, *HSF4*) genes in male left and right gonads, and female genital ridges. To determine whether there is any difference in the expression profile of heat treated and control groups, we pooled the RNAs of the individual samples group by group. As we found differences in the expression pattern compared to the heat treated and control groups, we performed qPCR runs from the individual samples to prove, whether these differences are statistically different or not. The individual chickens were chosen by random sampling. We compared the Delta Ct values of pooled samples with the average values of individual RNA samples in different groups. The number of used samples is indicated in Appendix A.

### 2.9. Statistical Analysis

To evaluate and analyse the collected data RStudio (1.0.136), R (R-3.2.2), GenEx (7.0) (MultiD Analyses AB, Göteborg, Sveden) and Excel (Microsoft Excel for Mac, 16.49 version) software were used. For the data obtained from the qPCR runs, expression changes of the target genes were calculated compared to the expression of the housekeeping gene with the standard 2^(−ΔΔCt) method, where Ct = cycle threshold; ΔCt = Ct (target gene)—Ct (housekeeping gene) and ΔΔCt = ΔCt (test sample)—ΔCt (control sample). The mean values in case of every group were compared using Welch’s t-tests. Prior testing, general assumptions of the t-test were checked, such as normality (Shapiro–Wilk’s test) and homogeneity of variances (Levene’s test). Furthermore, power analysis was made in case of every comparison to make sure the sample size was sufficient for testing. The categorical data was tested with Chi-squared tests. Significance levels were set as follows: * *p* < 0.05, ** *p* < 0.01, and *** *p* < 0.001.

## 3. Results

### 3.1. Analysis the Effect of Heat Stress on Reproductive Parameters of Heat-Treated Chickens

#### 3.1.1. Spermatological Analysis in Roosters

The spermatological parameters of heat-treated (HTHS) and control (HS) Transylvanian Naked Neck roosters were examined under heat stress. In the spermatological analysis, ten heat-treated and ten control roosters were used. Four parameters (quantity, concentration, motility and live–dead ratio) of sperm was determined (Figure 1). The volume of the semen was measured by pipetting. We could not find significant difference between the two experimental groups (*p* = 0.5075). According to the sperm concentration, no significant difference was found between the treated and the control groups (*p* = 0.1077) nor in the motility rate between the two groups (*p* = 0.6972). Finally, in the live–dead sperm ratio, no significant difference was found (*p* = 0.8816) between the two experimental groups. In summary, we can conclude that the pre-heat treatment has no effect on sperm quantity and quality.

#### 3.1.2. Examination the Egg Production and Fertilization Rate in Hens

In the females, two parameters, the egg production and the percent of unfertilized eggs were measured. The daily egg production of heat-treated chickens (HTHS) was significantly higher than the control (HS) hens in high environmental temperature (30 °C) (*p* = 0.00002) (Figure 2). Altogether, 1654 eggs were collected. The HTHS group produced 890 eggs, while in case of the HS group, we could collect 764 eggs. Based on candling at the 7th day of incubation in the case of HT group 55 eggs were discarded (6.18%), while in the HS group 42 eggs (5.42%) were discarded. The ratio of unfertilized eggs among the discarded ones was 10.91% in the treated and 42.86% in the control groups.

### 3.2. Comparison the Expression Profile of Heat-Shock Proteins and Heat-Shock Factors in Heat-Treated and Control Chicken Gonads

#### 3.2.1. Comparison the Delta Ct Values

The comparison showed high similarity in the expression profile of the pooled samples to the average of individual sample values (Appendix A). Comparing the Delta Ct values calculated from the individual samples, we found a significant difference between the chicks and adults (Figure 3(1A–6A)). A significant difference was determined between the Delta Ct values of *HSP70* (*p* = 0.0289) and *HSF3* (*p* = 0.0482) expressions in the heat-treated and control samples in case of adult male left gonads (Figure 3(1A) and Figure 3(5A)). In case of the right gonads, significant differences were found in *HSP70* (*p* = 0.023)*, HSF1* (*p* = 0.0007) and *HSF3* (*p* = 0.0013) (Figure 3(1B,3B); Figure 3(5B)) between the control chicks and control adults. Only the *HSP70* showed a significant difference comparing the values of control and heat-treated right gonads in adults (*p* = 0.0136) (Figure 3(2A)).

In case of the female gonads, only *HSF4* Delta Ct values defined a significant difference (*p* = 0.0016) between the control chicks and control adults (Figure 3(6C)). Significant differences were found between the control and heat-treated chicks in case of *HSP90* (*p* = 0.0355) (Figure 3(2C)) and *HSF4* (*p* = 0.0342) values (Figure 3(6C)). In other cases, we could not find significant differences between the groups.

The expression levels of 1: *HSP70* (*HSPA2*), 2: *HSP90* (*HSP90AA1*), 3: *HSF1*, 4: *HSF2*, 5: *HSF3* and 6: *HSF4* genes in chicks and adults, females-males, left and right gonads in case of HT and CTRL. A: Male left gonads, B: Male right gonads, C: Female gonads, HT: heat-treated, CTRL: control (* *p* < 0.05, ** *p* < 0.01 and *** *p* < 0.001).

#### 3.2.2. Comparison the Relative Expression Profile

To get more detailed information whether there is any significant difference in relative expression of HSPs and HSFs in control and treated samples, we calculated the relative expression values using GenEx (7.0) software (MultiD Analyses AB, Göteborg, Sveden). The *HSP90* (*p* = 0.0094) and *HSF4* (*p* = 0.0387) expression was significantly higher in the heat-treated female chick gonads than in the control (Figure 4E). However, in the adult female gonads all of the HSPs and HSFs decreased compared to the control groups, the *HSF2* (*p* = 0.0181) and the *HSF3* (*p* = 0.0011) were significantly lower in treated samples. (Figure 4F). In the case of chicks, there was no significant difference between the male left and right gonads compared to the control group (Figure 4A,C). However, in the left gonads of adults, the *HSP70* (*p* = 0.0002), *HSF1* (*p* = 0.0013), *HSF2* (*p* = 0.0217) and *HSF3* (0.0014) relative expression levels showed a significant increase compared to the controls (Figure 4D). Analysing the male right gonads in adult samples we found that the expression of all HSPs and HSFs increased. The expression of *HSP70* (*p* = 0.0052), *HSF1* (*p* = 0.0333), *HSF3* (*p* = 0.0332) and *HSF4* (0.0498) showed a significant increase compared to the control (Figure 4B).

## 4. Discussion

El-Tarabany [40] reported the impact of high temperature humidity index in Japanese quail. They found that the control groups had significantly greater fertility and hatchability than the heat stressed group. It was published that in broiler chickens the genetically lean breed is more resistant to the higher ambient temperature, than the fat counterpart [20]. Laine and her colleagues [41] studied the effect of higher temperature in the hypothalamic–pituitary–gonadal–liver axis of Great Tit (*Parus major*). They found that the zona pellucida glycoprotein 4 (ZP4) is differently expressed before and after the onset of egg-laying [41]. On the other hand, we could not detect difference in spermatological parameters between the heat-treated and control groups under heat-stress (Figure 1). Analysing the female reproductive performance, we found significantly higher egg production and fertility rate (Figure 2). However, Végi and her colleagues found that the spermatological parameters decline after heat-treatment in Cobb chicken hybrid [42]. It was published that the Naked Neck chickens show better body temperature regulation and higher radiation rates from the naked neck than the covered neck breeds if they are kept at 35 °C [43].The reason why we did not find any difference among the spermatological parameters might be the effect of the Transylvanian Naked Neck breed

Mezquita et al. [44] investigated the *HSP70* expression in adult chicken testes. They found that the HSP70 was highly expressed in the left and right testes at higher temperatures (44 or 46 °C). However, at normal internal temperature the *HSP70* is not expressed in the left gonad but it is present in the regressed (right) gonad [44]. We could detect low *HSP70* expression level, but we found significantly higher *HSP70* expression at adult age in roosters both in left and right gonads when they were heat treated. Interestingly, we found that *HSF3* expression level increased parallel with *HSP70* expression. Zhang and his colleagues [10] found that the level of HSP70 declined in the heart 6 hours after the heat stress, but the HSF3 expression remained high. Tarkhan and his colleagues [45] examined the *HSP70* and *HSF3* expression levels in cold stress. They found decreased expression levels in the liver in case of both genes.

In AA Broiler breed (from China) the *HSP90* mRNA level increased in the liver, heart and kidney after 2 hours of high temperature. The *HSP90* expressed in the endothelium cells and the blood vessel walls, which influences the regulation of the blood flow [46]. Hao and Gu examined the expression of *HSP90* on *pectoralis major* in broiler breed after acute heat stress. They found that the *HSP90* expression is positively correlate with corticosterone and superoxide dismutase, but negatively correlate with the pH in *pectoralis major* [47].

The gonads are the place where gametes are produced, so it appears useful to compare the impact of stress on both the tissues and the cells that result from it. In order to start the mechanistic studies, it is first necessary to identify the actors who intervene and the first screening that was done made it possible to find actors who are sensitive. This is the first step to launch mechanistic approaches.

We found high *HSP90* expression in chickens compared to the *HSP70* expression level. We could observe significant differences only in case of female chicks between the control and heat-treated *HSP90* expression level.

It was reported that numerous transcripts in the testes expressed differentially between the heat-stressed broiler-type and layer-type chickens [48]. We found in the left gonads of the adult heat-treated males that the *HSP70*, *HSF1*, *HSF2* and *HSF3* relative expression levels showed significant increase compared to the controls. Whether these expression patterns associate with the heat-tolerance require a further investigation. It was found that after 2 hours of heat treatment the expression of *HSP27*, *HSP90* and *HSP70* increased in a Taiwanese country chicken rooster, but the mRNA of *CDH5*, *CIRBP*, *SLA* and *NTF3* were downregulated in the testes [48]. Wang et al. [49] published that in the heat-stressed chicken testes the proteins that involve in autophagy and the major HSPs (HSP90α, HSPA5, HSPA8) were upregulated but the proteins that negatively regulate apoptosis were downregulated. In the future, we plan to check the expression level of these factors in our heat-treated samples, too.

Furukawa et al. [50] shows that the *HSF1* is a very important regulator in the ovarian differentiation of Medaka. They made an HSF1 knock-out animal and found that HSF1 protects the female germ cells under heat stress. We could detect significantly higher *HSF1* expression in heat-treated roosters, in both left and right gonads, compared to the control, but there was no significant difference in the level of *HSF1* in treated and control females.

HSF2 is very important in the development of brain and reproductive organs, but the fundamental rule is not identified yet [51]. In *HSF2* knock-out B Lymphocyte cells they found that the KO line was more sensitive to the heat stress than the wild type [52]. We found higher *HSF2* expression in heat-treated gonads in adult roosters, but significantly lower expression in adult females.

The mutation of *HSF4* gene may cause a congenital or senile cataract in human. We found significantly higher *HSF4* expression in heat-treated female chicks parallel with high HSP90 expression. In case of males, we could not detect difference in the expression profile between the heat-treated and the control ones. According to these findings, we propose that the increased *HSP90* and *HSF4* levels could eliminate somehow the effect of heat stress, but further analysis is needed to find the molecular pathways responsible for this effect.

## 5. Conclusions

The average global temperature has increased over the century. Heat-shock proteins and heat-shock factors play an essential role in normal cellular physiology and protection against different stressors, including heat stress. In chickens, HSP and HSF levels are increased in almost all the tissues in response to heat stress. This increased HSP level protects cellular proteins from heat-stress induced damage. We found that the post-hatch (24 h after hatching) heat manipulation had an influence mainly on the female reproductive parameters, while in adult animals, we found significant difference in heat-stress protein and heat-shock factor expressions in both genders. These are indeed the first hypotheses and only future experiments of genetic modifications of these actors will be able to validate these hypotheses. With the difficulty of generating animals in a reasonable time scale.

The search for genetic variants (SNP for example) more particularly in these genes would also be an approach to be pursued in order to try to correlate the character with the impact of the genes. Our findings show a significant effect on egg production but not on the sperm quality after post-hatch heat treatment. The egg production is more complex, longer and energy intense process than spermiogenesis and that could be one of the reasons why we could not find any difference in the sperm quality between the control and heat-stressed group. The found significant differences might be related to the increased expression level of *HSP90* and *HSF4* in heat-treated female chicks.

## Figures and Tables

**Figure 1 animals-11-01575-f001:**
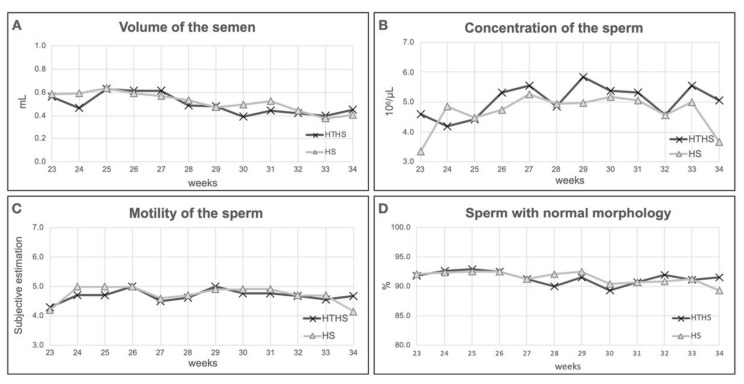
In this figure the results of heat-treated (HTHS) and control (HS) groups under 12 weeks of heat stress on sperm parameters are summarized. Significant difference between the two groups was not found. (**A**): Volume of the semen. (**B**): Concentration of the sperm. (**C**): Motility of the sperm. (**D**): Sperm with normal morphology.

**Figure 2 animals-11-01575-f002:**
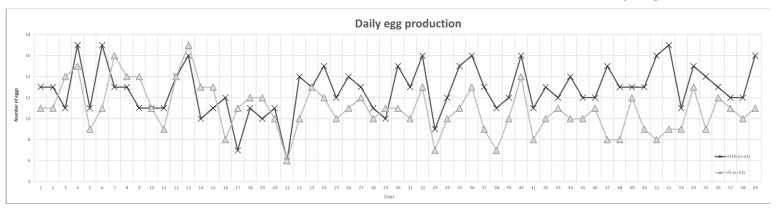
Demonstration of the daily egg production and the ratio of fertilized eggs in heat-treated (HTHS) and control (HS) groups under heat stress. The egg collection began from the 24th life week but the data in the figure presents the egg number from the 27th life week because the egg production was stable from this time.

**Figure 3 animals-11-01575-f003:**
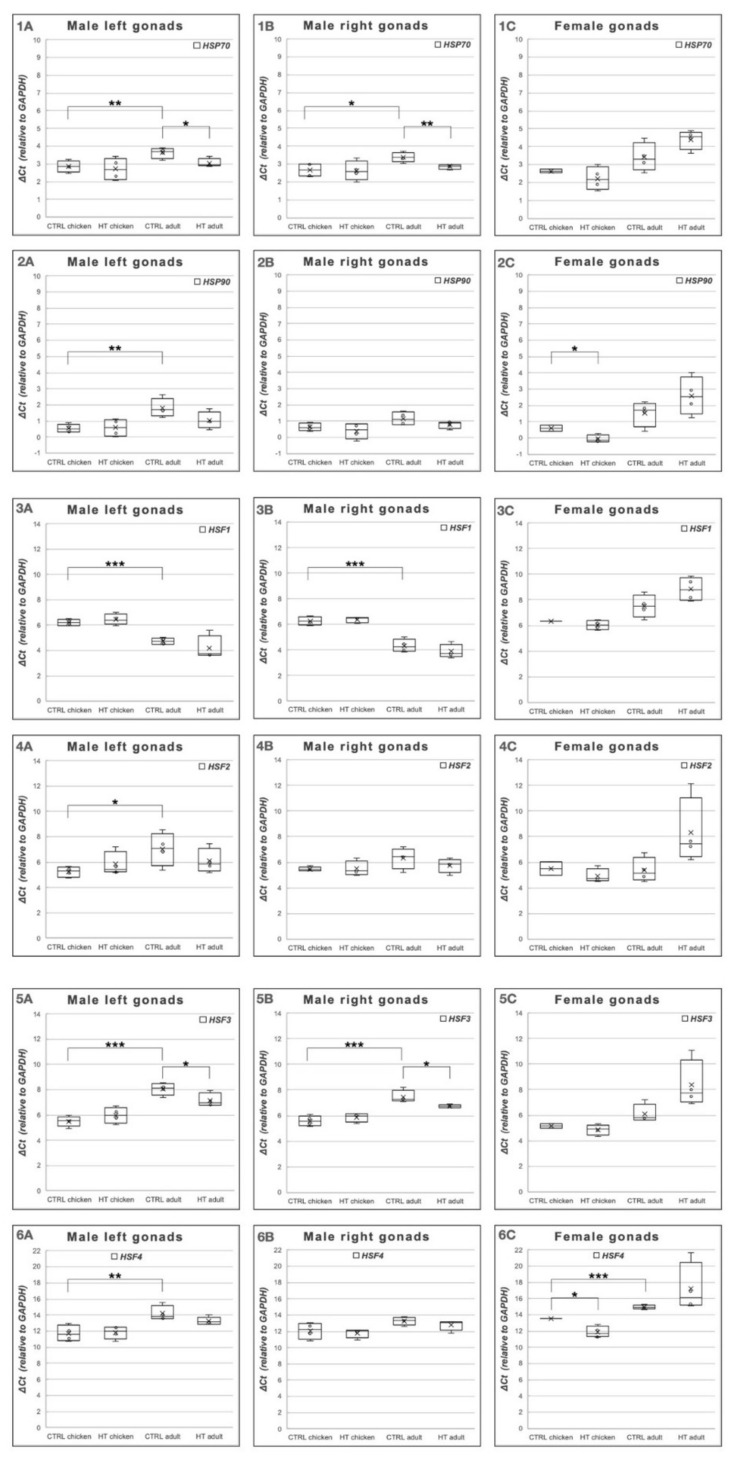
Delta Ct values of two heat-shock protein (*HSP70* and *HSP90*) and four heat-shock factor *HSF1*, *HSF2*, *HSF3*, *HSF4*) genes were determined in male and female left gonads of (heat-treated) HT and (control) CTRL chicks and adults. In every case the *GAPDH* was the reference gene. (**1**): HSP70; (**2**): HSP90; (**3**): HSF1; (**4**): HSF2; (**5**): HSF3; (**6**): HSF4. (**A**): Male left gonads; (**B**): Male right gonads; (**C**): Female gonads. * *p* < 0.05, ** *p* < 0.01, *** *p* < 0.001.

**Figure 4 animals-11-01575-f004:**
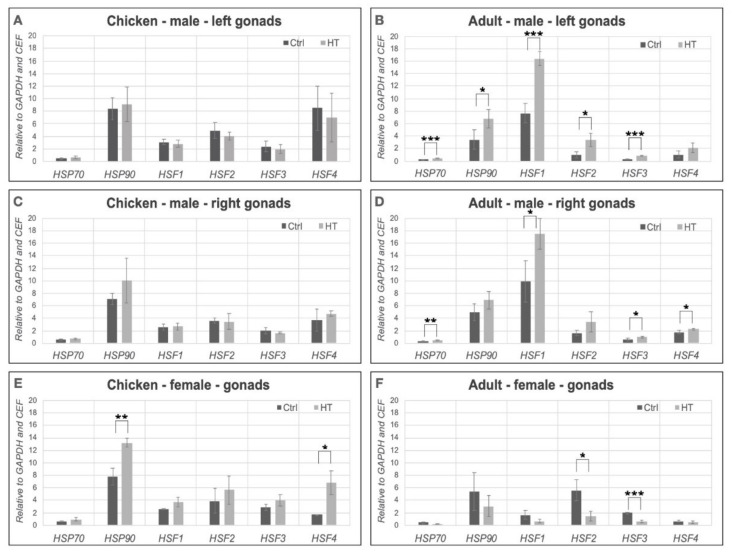
The bar charts show the relative expression values in female and male chick and adult samples in control and heat-treated gonads relative to chicken embryonic fibroblast sample. The relative expression values of heat-shock protein (*HSP70* and *HSP90*) and heat-shock factor (*HSF1, HSF2, HSF3, HSF4)* genes were determined in female and male gonads. *GAPDH* was chosen as reference gene. (**A**): Chicken mal left gonads. (**B**): Adult male left gonads. (**C**): Chicken male right gonads. (**D**): Adult male right gonads. (**E**): Chicken female gonads. (**F**): Adult female gonads. * *p* < 0.05, ** *p* < 0.01, *** *p* < 0.001.

## Data Availability

The data presented in this study are available on request from the corresponding author.

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
