# Peer review of "Effect of Post-Hatch Heat-Treatment in Heat-Stressed Transylvanian Naked Neck Chicken"

_animals, 2021, doi:10.3390/ani11061575_

Round 1

Reviewer 1 Report

  1. The heat stress treatment test was not clearly.For the first 24 hours after hatching, the chicks were placed under an infrared lamp at 32°C on absorbent paper litter. Then the chicks were heat-treated at 38.5°C ambient temperature with 60% humidity for 12 hours. How about the control group?
  2. Inadequate analysis of mechanisms. The results consist of two main parts. The effect of heat stress on semen quality of roosters and egg production and fertilization rate of hens. Comparison the expression profile of heat shock proteins and heat shock factors in heat- treated and control chicken gonads. These two parts of the work involve very little work on mechanisms.
  3. The authors speculate in the discussion sectionthat significant effect on egg production might be related to the increased expression level of HSP90 and HSF4 in heat-treated female chickens. But this needs more evidence.

Author Response

Thank you very much for your suggestions. We corrected in the text the mistakes and we responded to your questions.

  • The heat stress treatment test was not clearly. For the first 24 hours after hatching, the chicks were placed under an infrared lamp at 32°C on absorbent paper litter. Then the chicks were heat-treated at 38.5°C ambient temperature with 60% humidity for 12 hours. How about the control group?

160 chicks were hatched on a regular way (37.5°C and 60% humidity). Then 80 chicks were placed back to the hatcher for heat treatment. The temperature was set to 38.5°C and the humidity was 60% for 12 hours. The control 80 chicks were kept at 32°C.

It was clarified in the text.

  • Inadequate analysis of mechanisms. The results consist of two main parts. The effect of heat stress on semen quality of roosters and egg production and fertilization rate of hens. Comparison the expression profile of heat shock proteins and heat shock factors in heat- treated and control chicken gonads. These two parts of the work involve very little work on mechanisms.

The gonads are the place where gametes are produced, so it appears useful to compare the impact of stress on both the tissues and the cells that result from it. In order to start the mechanistic studies, it is first necessary to identify the actors who intervene and the first screening which was done made it possible to find actors who are sensitive. This is the first step to launch mechanistic approaches.

  • The authors speculate in the discussion section that significant effect on egg production might be related to the increased expression level of HSP90 and HSF4 in heat-treated female chickens. But this needs more evidence.

These are indeed the first hypotheses and only future experiments of genetic modifications of these actors will be able to validate these hypotheses. With the difficulty of generating animals in a reasonable time scale. The search for genetic variants (SNP for example) more particularly in these genes would also be an approach to be pursued in order to try to correlate the character with the impact of the genes.

Reviewer 2 Report

The author performed a post-hatch heat-treatment followed by heat-stress (HTHS) experiment on Transylvanian Naked Neck chickens where gonadal performance was measured. Some HSF/HSP expression were measured following HT on 1-day old hatchlings and adult birds from the end of the experiment. Unfortunately, I was not able to follow a crucial part of the experimental design based on the current description in Method for heat stress (P3, L125-133). What was the length of the elevated temperature exposure? In the abstract (P1, L30-32) it was suggested that the heat stress was for 2 weeks. However, performance data was presented well beyond 2 weeks. If the heat stress was discontinued after 2 weeks, then the results from this experiment becomes quite tricky to interpret as this is not the typical heat stress protocol. I sincerely hope the study did not have such a fatal design flaw and the stated heat stress treatment timeline in the Abstract was just a mistake. Please see below for some other suggestions for minor revision to improve the manuscript (if there isn't a fatal design flaw to the HTHS piece):

P3, L122-124: How many animals went into Experiment I and II?

P3, L125-133: Please describe more clearly the heat stress applied. What age were the birds when the heat stress started? How long was the heat stress? Were all 160 birds subjected to heat stress?

P3, L134-139: When did the egg collection begin? Figure 2a only shows "days" starting at 1, which is not informative about the age of the laying hens.

P5, L196-199: What samples were used to test the housekeeping genes?  Why were the n different between SF1a and SF1b?

P5, L205-209: How were the individual samples chosen from the pool?

P5, L211-212: What happened to ST1a? Why are the Supplementary Tables presented out of order with ST2 appearing on P4, L189? Also, "HS" and "HTHS" was not defined in ST1 description. Lastly, is the description "Ctrl" and "HT" appropriate for the Adult birds? If all 160 birds were heat stressed in Experiment I, wouldn't the Adults be "HS" and "HTHS"?

In general, it was confusing that the term "chicken" vs. "adult", did the authors meant "chicks"? This unclear use of terminology first appeared in ST1.

P4-5, L180-212: How were the qPCR ran? More specifically, was everything done a single plate? If not, was the housekeeping gene used on every plate? Please clarify as this may have an impact toward your statistical modeling in GeneEx analysis later.

P5, L213-222: Please provide more detailed description on the statistical analysis performed in GeneEx. Specifically, were plate effect corrected? How was relative expression calculated? Was a particular statistical model used?

P6, L248-249: Why were different number of eggs analyzed for embryonic development from the HTHS (n=108) and HS (n=172) group? Also, please double check % presented in Figure 2c. 104/257 = 40.4% and 153/257 = 59.5%.

P7, L262: Is there an additional Supplemental Figure? Because SF1 was showing housekeeping genes.

P9, L280-283: Should be formatted to continue from Figure 3.

P9, L284-298: Text uses "Chicken" and "Chicks" interchangeably. Made this section very confusing to read. See comment above about these terms.

P9, Figure 4: Based on the description for the Y-axis, it seems that the relative expression was normalized against CEF's GAPDH expression. This is where a more detailed description of the qPCR design and GeneEx will be helpful. Currently, it difficult to accept the statistics without knowing if the normalization was done properly.

P10, L316: Statistics were not shown for fertility rate.

P10-11, L324-373: This is where the heat-stress design is super crucial. A lot of references to other heat-stress experiments were cited, but if the design of this study was done incorrectly, the entire discussion about gene expression will need to be reworked.

Author Response

Thank you very much for your suggestions. We corrected in the text the mistakes and we responded to your questions.

  • The author performed a post-hatch heat-treatment followed by heat-stress (HTHS) experiment on Transylvanian Naked Neck chickens where gonadal performance was measured. Some HSF/HSP expression were measured following HT on 1-day old hatchlings and adult birds from the end of the experiment. Unfortunately, I was not able to follow a crucial part of the experimental design based on the current description in Method for heat stress (P3, L125-133). What was the length of the elevated temperature exposure? In the abstract (P1, L30-32) it was suggested that the heat stress was for 2 weeks. However, performance data was presented well beyond 2 weeks. If the heat stress was discontinued after 2 weeks, then the results from this experiment becomes quite tricky to interpret as this is not the typical heat stress protocol. I sincerely hope the study did not have such a fatal design flaw and the stated heat stress treatment timeline in the Abstract was just a mistake. Please see below for some other suggestions for minor revision to improve the manuscript (if there isn't a fatal design flaw to the HTHS piece):

Thank you, the remark. Yes, the heat stress treatment timeline in the Abstract was just a mistake. The duration was 12 weeks. It was corrected in the text.

  • P3, L122-124: How many animals went into Experiment I and II?

Experiment I: 80 control and 80 treated animals were raised up together and then 31-31 females and 3-3 males went under heat stress, plus 10-10 roosters in individual cages in the same air space.

It was clarified in the text.

Experiment II: 15 control and 15 treated animals. It was clarified in the text.

  • P3, L125-133: Please describe more clearly the heat stress applied. What age were the birds when the heat stress started? How long was the heat stress? Were all 160 birds subjected to heat stress?

In Experiment I. 80-80 animals were in the treated and control groups. We randomly selected for the further reproductive biology examinations under heat stress 31-31 layers with 3-3 males in the 24th life week. The average temperature in the pen at the height of the birds' habitat was constantly 30°C during 12 weeks for all treated and control animals.

It was clarified in the text.

  • P3, L134-139: When did the egg collection begin? Figure 2a only shows "days" starting at 1, which is not informative about the age of the laying hens.

The egg collection began from 24th life week. It was corrected in Figure 2a.

  • P5, L196-199: What samples were used to test the housekeeping genes?  Why were the n different between SF1a and SF1b?

We tested the expression of two housekeeping genes (GAPDH and ß-Actin). We compared the average Ct values of GAPDH and ß-Actin in control and heat-treated left gonad ‘s using pooled RNA samples. We found lower Ct values in control and heat treated and lower standard deviation in control samples using GAPDH (Supplementary Figure 1A). There weren’t significant differences comparing the ß-Actin Ct values in control and heat-treated samples (p=0.8454) and at GAPDH values in control and heat-treated samples (p=0.4764).

We modified the Supplementary Figure 1a switching Row/Column values, we hope that in this form the comparison is more apparent.

In Supplementary Figure 1b we compared all of the examined samples (left gonads of males and females, and right gonads of males). There was no significant difference between the control and heat-treated samples (p= 0.523) comparing the average GAPDH Ct values.

In Supplementary Figure 1b the average Ct and SD values of the CEF are presented (n=8, as we analysed 8 plates, and on each plates, we used CEF as plate reference). As the difference was very low, we could compare the results.

  • P5, L205-209: How were the individual samples chosen from the pool?

The individual samples were chosen by random sampling. In the chick female control group, we had only 2 individuals.

  • P5, L211-212: What happened to ST1a? Why are the Supplementary Tables presented out of order with ST2 appearing on P4, L189? Also, "HS" and "HTHS" was not defined in ST1 description. Lastly, is the description "Ctrl" and "HT" appropriate for the Adult birds? If all 160 birds were heat stressed in Experiment I, wouldn't the Adults be "HS" and "HTHS"?

We checked the supplementary tables, and we corrected the names of the group in different experiments.

  • In general, it was confusing that the term "chicken" vs. "adult", did the authors meant "chicks"? This unclear use of terminology first appeared in ST1.

We changed all of “chicken” words to “chick”, where we meant the young chicken in the tables and the text too.

  • P4-5, L180-212: How were the qPCR ran? More specifically, was everything done a single plate? If not, was the housekeeping gene used on every plate? Please clarify as this may have an impact toward your statistical modelling in GeneEx analysis later.

We used chicken embryonic fibroblast as a reference sample at GenEx analysis and as plate quality control. We placed into each plates the same chicken embryonic fibroblast derived cDNA. At all analysis we used the same threshold. As quality control, before the qPCR analysis we checked the GAPDH Ct values of the reference CEF sample

  • P5, L213-222: Please provide more detailed description on the statistical analysis performed in GeneEx. Specifically, were plate effect corrected? How was relative expression calculated? Was a particular statistical model used?

The real-time PCR was performed using Eppendorf Realplex 4 qPCR Real Time Cycler. The Eppendorf Real Time program provided by the company made the basic quality control, and calculated the Ct values for GenEx analysis. For the data analysis obtained from the qPCR runs, we used GenEx qPCR analysis software. Expression changes of the target genes were calculated compared to the expression of the housekeeping gene with the standard 2^(−ΔΔCt) method, where Ct = cycle threshold; ΔCt = Ct (target gene) − Ct (housekeeping gene) and ΔΔCt = ΔCt (test sample) − ΔCt (control sample). It was used t-test were checked, such as normality (Shapiro-Wilk’s test) and homogeneity of variances (Levene’s test).

  • P6, L248-249: Why were different number of eggs analyzed for embryonic development from the HTHS (n=108) and HS (n=172) group? Also, please double check % presented in Figure 2c. 104/257 = 40.4% and 153/257 = 59.5%.

Thank you for the suggestions. We checked again the results and corrected the text.

  • P7, L262: Is there an additional Supplemental Figure? Because SF1 was showing housekeeping genes.

We have not other Supplementary Figures.

  • P9, L280-283: Should be formatted to continue from Figure 3.

We formatted the Figure 3. in the text.

  • P9, L284-298: Text uses "Chicken" and "Chicks" interchangeably. Made this section very confusing to read. See comment above about these terms.

We changed the “chicken” to “chick”, where we mean the young chicken in the tables and the text too.

  • P9, Figure 4: Based on the description for the Y-axis, it seems that the relative expression was normalized against CEF's GAPDH expression. This is where a more detailed description of the qPCR design and GeneEx will be helpful. Currently, it difficult to accept the statistics without knowing if the normalization was done properly.

The mean values in case of every group were compared using Welch’s t-tests. Prior testing, general assumptions of the t-test were checked, such as normality (Shapiro-Wilk’s test) and homogeneity of variances (Levene’s test). Furthermore, power analysis was made in case of every comparison to make sure the sample size was sufficient for testing. The categorical data was tested with Chi-squared tests.

  • P10, L316: Statistics were not shown for fertility rate.

We made a statistical analysis for fertility rate. We put the appropriate place in text.

  • P10-11, L324-373: This is where the heat-stress design is super crucial. A lot of references to other heat-stress experiments were cited, but if the design of this study was done incorrectly, the entire discussion about gene expression will need to be reworked.

The heat stress treatment timeline in the Abstract was just a mistake. The duration was 12 weeks. It was corrected in the text.

Reviewer 3 Report

The objective of this paper is to measure the impacts of an early heat treatment (1st day after hatching) on 1) the expression of temperature sensitive factors in the gonads just after the treatment and 2) the reproductive parameters of heat stressed adult males and females.

Despite the interest of these issues, the paper requires major modifications to be publishable.

General comments:

In general, it would be more interesting to focus the literature review (introduction and discussion) on the impact of heat treatment on the reproductive system (and not on the lens, or thermoregulation, or even cataract!...) to better understand the purpose of the study. References on this subject are missing.

Overall, the discussion does not allow for any concrete assumptions. No link is made between the two experiments in this study, (between the results of the direct impact of the treatment and the later results on the reproductive organs), although this would be very interesting.

It would be more logical to reverse the order of the experiments (first the expression study just after the heat treatment and then the impact on the reproductive parameters when these animals have become adults).

Comment or Minor revision suggested:

Title: 

The title is misleading because the heat-stress is a challenge after heat-treatment programming. They are looking at the impact of the treatment on the response to a heat-stress... Moreover it might be good to specify that the results are mainly focused on the parameters of reproduction.

Simple summary and abstract : 
In the second experiment, it must be specified that the RNA samples are taken directly after the heat treatment.

Introduction: 
The introduction could be easily shortened by focusing only on the information essential to understanding the experiments undertaken in this study. The references are too detailed, for example what is the point of talking about the development of the lens in mammals (line 100) or in zebrafish (line28) ?

Line 52: Is it the rise in internal temperature (fever) that alters the functions mentioned below, or is it the environmental temperature? the succession of the two sentences leaves a doubt…

Line 88: “Under stress condition in poultries all heat shock factors are activated”. This sentence should be moderated or supported by references...

The second-last paragraph is not very clear... : Line101: what does “The transition during heat-stress » mean?

Materials and methods: 

Line 124: DNA and RNA analysis 

2.1 Heat treatment:
During the heat treatment, does the control group stay under the lamp at 32°C? 
2.3 Heat stress:
The conditions of heat stress are not sufficiently detailed: what is its duration? under what conditions are the controls maintained?

2.4 embryonic abnormality:
It should be specified that these are the eggs of the females of experiment 1, for the measurement of egg laying quality.
What does "Every week 20-20 eggs were candled on the 7th day of incubation" mean?
What does “The other eggs were incubated the regular way” mean ? Was there any heat treatment during the incubation of the eggs?

2.6 collection of gonadal tissue:
Was there an ethical evaluation for this study? Is euthanasia by cervical dislocation well adapted to the age of the animals (method reserved for birds weighing less than 1kg according to the European directive)? Specify the experience (I. or II.) concerned in this paragraph

2.8 Real time PCR:
I understand the choice of the reference gene GAPDH in relation to the lower variability than actin regarding CT values. But on supplemental figure1b, why the number of samples is not the same as on 1a?

If you are using the CEF as a reference sample, shouldn't you also test whether there is a significant difference with each of your two experimental groups (concerning the Ct average values) ?

What do you call delta CT (CTreference -CTsample, or CTtarget gene-CTreference gene) ? Why don't you measure the relative expression (as explained in the statistical analysis paragraph) ?

2.9 statistical analysis
In the supplementary table 1b2 it is stated that the number of samples for RNA does not exceed 4, so it is not possible to compare the values with a T test.

Results: 

3.1.1
Did the heat stress cause an alteration of the reproductive parameters? It would have been interesting to compare with the control groups, which were not treated by heat stress.

3.1.2
Heat stress is applied for 2 weeks according to the abstract (line31), but the age of the animals at the time of stress is not specified (neither in the abstract, nor in the intro nor in the material and method).

In this part of the results, eggs are collected for 59 days. Could you please specify the period of time during which the females were under heat stress condition?

How did you measure the statistical difference between females in their daily egg production? In the materials and methods you talk about t test, but did you compare the total eggs laid over the whole period or each day?

Line 250: “we opened the eggs”, do you mean the incubator to make the mirage instead? The presence of an embryo (thus fertility) does not require to open the eggs. 
Where are the results concerning the measures of abnormality of the embryos (which require this time to open the eggs)

In the materials and methods it says that the eggs are mirrored on the 7th day... Why are you talking about the 3rd day here?

3.2.1
Line 262- sup. Fig. 2

I don't understand what the comparison of Delta Ct values brings... What is the additional information compared to the relative expression?

Discussion:

Explanations for the lack of effect on sperm qualities are unconvincing. What are the differences between the experience of this study and the cited reference showing that there is an impact of heat treatment in the same breed (ref 45) ?

Line 370: according to what findings do you propose that HSF4 and HSP90 could eliminate the effect of heat stress? The logic is really not clear…

In the abstract it is said that there might be a link between egg production and the expression level of HSF4 and HSP90. This part is not covered in the discussion

Author Response

Thank you very much for your suggestions. We corrected in the text the mistakes and we responded to your questions.

  • The objective of this paper is to measure the impacts of an early heat treatment (1st day after hatching) on 1) the expression of temperature sensitive factors in the gonads just after the treatment and 2) the reproductive parameters of heat stressed adult males and females.

Despite the interest of these issues, the paper requires major modifications to be publishable.

General comments:

  • In general, it would be more interesting to focus the literature review (introduction and discussion) on the impact of heat treatment on the reproductive system (Roland) (and not on the lens, or thermoregulation, or even cataract!...) to better understand the purpose of the study. References on this subject are missing.

We integrate a new reference about the effect of heat stress on the ovarian function of White Leghorn laying hens.

  • Overall, the discussion does not allow for any concrete assumptions. No link is made between the two experiments in this study, between the results of the direct impact of the treatment and the later results on the reproductive organs), although this would be very interesting.

These are indeed the first hypotheses and only future experiments of genetic modifications of these actors will be able to validate these hypotheses. With the difficulty of generating animals in a reasonable time scale. The search for genetic variants (SNP for example) more particularly in these genes would also be an approach to be pursued in order to try to correlate the character with the impact of the genes.

  • It would be more logical to reverse the order of the experiments (first the expression study just after the heat treatment and then the impact on the reproductive parameters when these animals have become adults).

At Experiment I. we wanted to see whether the higher temperature has any effect on the productivity traits. In Experiment II. we repeated the heat treatment and checked the gene expression profile in reproductive organs immediately after the treatment. We presented the experiment in chronological order.

Comment or Minor revision suggested:

Title: 

  • The title is misleading because the heat-stress is a challenge after heat-treatment programming. They are looking at the impact of the treatment on the response to a heat-stress... Moreover it might be good to specify that the results are mainly focused on the parameters of reproduction.

We modified the article title.

Simple summary and abstract:

  • In the second experiment, it must be specified that the RNA samples are taken directly after the heat treatment.

We amended the simple summary section with the RNA sample collection. details.

Introduction:

  • The introduction could be easily shortened by focusing only on the information essential to understanding the experiments undertaken in this study. The references are too detailed, for example what is the point of talking about the development of the lens in mammals (line 100) or in zebrafish (line28) ?

Thank you for your comment. We deleted the mammalian lens and one of the zebrafish references. We kept that article where zebrafish HSF5 was discovered because the HSF5 was a new member of the heat shock factor family.

  • Line 52: Is it the rise in internal temperature (fever) that alters the functions mentioned below, or is it the environmental temperature? the succession of the two sentences leaves a doubt…

Yes, in the first sentence we forgot to mention that is environmental temperature and not internal temperature. We corrected in the text.

  • Line 88: “Under stress condition in poultries all heat shock factors are activated”. This sentence should be moderated or supported by references...

We moderated in the text

  • The second-last paragraph is not very clear... : Line101: what does “The transition during heat-stress » mean?

Thank you, we corrected that sentence.

Materials and methods:

  • Line 124: DNA and RNA analysis

We corrected in the text. We changed the DNA to RNA.

  • 2.1 Heat treatment:
    During the heat treatment, does the control group stay under the lamp at 32°C? 

Yes, the control group stayed at 32°C under the lamp.

  • 2.3 Heat stress:
    The conditions of heat stress are not sufficiently detailed: what is its duration? under what conditions are the controls maintained?

The heat stress was applied both for control and treated groups, from 24th life week during 12 weeks. It was corrected in the text.

  • 2.4 embryonic abnormality:
    It should be specified that these are the eggs of the females of experiment 1, for the measurement of egg laying quality.
    What does "Every week 20-20 eggs were candled on the 7th day of incubation" mean?
    What does “The other eggs were incubated the regular way” mean ? Was there any heat treatment during the incubation of the eggs?

The corrected text, which gives answer for your questions, is the following:

In Experiment I eggs were collected daily from the 24th life week and marked with group number and date and placed to the incubator once a week. Every week 20-20 eggs were applied for molecular biological examinations. The incubation was made on regular way (37.5°C and 60% humidity). Eggs were candled on the 7th day of incubation. The ratio of fertile and infertile eggs was determined.

  • 2.6 collection of gonadal tissue:
    Was there an ethical evaluation for this study? Is euthanasia by cervical dislocation well adapted to the age of the animals (method reserved for birds weighing less than 1kg according to the European directive)? Specify the experience (I. or II.) concerned in this paragraph

Cervical dislocation was applied in Experiment II for 2 days old chicks. It was corrected in the text.

  • 2.8 Real time PCR:
    I understand the choice of the reference gene GAPDH in relation to the lower variability than actin regarding CT values. But on supplemental figure1b, why the number of samples is not the same as on 1a?

We tested the expression of two housekeeping genes (GAPDH and ß-Actin). First we compared the average Ct values of GAPDH and ß-Actin only in control and heat-treated left gonad ‘s using pooled RNA samples (Supplementary figure 1a.). In supplementary figure 1b we compared all of the examined samples, left gonads of males and females, and right gonads of males.

  • If you are using the CEF as a reference sample, shouldn't you also test whether there is a significant difference with each of your two experimental groups (concerning the Ct average values)?

There weren’t significant differences comparing the ß-Actin Ct values in control and heat-treated samples (p=0.8454) and at GAPDH values in control and heat-treated samples (p=0.4764).

  • What do you call delta CT (CTreference -CTsample, or CTtarget gene-CTreference gene) ? Why don't you measure the relative expression (as explained in the statistical analysis paragraph) ?

Delta CT was calculated CT target gene - CT reference gene. First, we made the statistical analysis comparing each group (male-female, left-right, young-adult using delta Ct values). After that we made the comparison of heat-treated and non-treated samples using the relative expression values.

  • 2.9 statistical analysis
    In the supplementary table 1b2 it is stated that the number of samples for RNA does not exceed 4, so it is not possible to compare the values with a T test.

There are assumptions have to be met before performing a Welch’s t-test, and sample size is definitely an important factor influencing the power of a test, however it is not a strict condition. Indeed, we had 4 samples/group, but this alone is not a reason for exclusion of the t-test. Smaller sample size is also acceptable if conditions are met. Prior testing, general assumptions of the t-test were checked, such as outliers, normality (Shapiro-Wilk’s test) and homogeneity of variances (Levene’s test). We found that our data in the compared groups are normally distributed and variances are equal. Furthermore, power analysis was made in case of every comparison to make sure the sample size was sufficient for such testing.

Results:

  • 3.1.1
    Did the heat stress cause an alteration of the reproductive parameters? It would have been interesting to compare with the control groups, which were not treated by heat stress.

Each group was examined under heat stress.

  • 3.1.2
    Heat stress is applied for 2 weeks according to the abstract (line31), but the age of the animals at the time of stress is not specified (neither in the abstract, nor in the intro nor in the material and method).

In this part of the results, eggs are collected for 59 days. Could you please specify the period of time during which the females were under heat stress condition?

The heat stress duration was 12 weeks. It was corrected in the Abstract and in the Material and methods part.

  • How did you measure the statistical difference between females in their daily egg production? In the materials and methods, you talk about t test, but did you compare the total eggs laid over the whole period or each day?

It was compared the total eggs laid over the whole period.

  • Line 250: “we opened the eggs”, do you mean the incubator to make the mirage instead? The presence of an embryo (thus fertility) does not require to open the eggs. 
    Where are the results concerning the measures of abnormality of the embryos (which require this time to open the eggs)?

Sometimes you can find embryos died early stages of those eggs, which seems unfertile at candling as well as it is possible to determine different embryonic death fenotypes. That’s why we opened those eggs which were discarded at candling. These results were not showed in our manuscript so we modified this part in the Material and methods

  • In the materials and methods, it says that the eggs are mirrored on the 7th day... Why are you talking about the 3rd day here?

It was 7th day. It is corrected in the text.

  • 3.2.1
    Line 262- sup. Fig. 2

I don't understand what the comparison of Delta Ct values brings... What is the additional information compared to the relative expression?

Delta CT was calculated CT target gene - CT reference gene. First, we made the statistical analysis comparing each group (male-female, left-right, young-adult using delta Ct values). After that we made the comparison of heat-treated and non-treated samples using the relative expression values.

Discussion:

  • Explanations for the lack of effect on sperm qualities are unconvincing. What are the differences between the experience of this study and the cited reference showing that there is an impact of heat treatment in the same breed (ref 45) ?

It was not the same breed. It is corrected in the text:

Analysing the female reproductive performance, we found significantly higher egg production and fertility rate (Fig.2). However, Végi and her colleagues found that the spermatological parameters decline after heat-treatment in Cobb  chicken hybrid [45]. It was published that the Naked Neck chickens show better body temperature regulation and higher radiation rates from the naked neck than the covered neck breeds if they are kept in 35 oC [46].The reason why we did not find any difference among the spermatological parameters might be the effect of the Transylvanian Naked Neck breed.

  • Line 370: according to what findings do you propose that HSF4 and HSP90 could eliminate the effect of heat stress? The logic is really not clear… In the abstract it is said that there might be a link between egg production and the expression level of HSF4 and HSP90. This part is not covered in the discussion

These are indeed the first hypotheses and only future experiments of genetic modifications of these actors will be able to validate these hypotheses. With the difficulty of generating animals in a reasonable time scale.